# 'It felt like a black hole, great uncertainty, but we have to take care for our patients'– Qualitative findings on the effects of the COVID-19 pandemic on specialist palliative home care

**Maximiliane Jansky**[1][ID][☯]*, **Franziska Schade**[1][☯], **Nicola Rieder**[1], **Danica Lohrmann**[1], **Cordula Gebel**[ID][2], **Lars Kloppenburg**[2], **Ulrich Wedding**[2], **Steffen T. Simon**[3], **Claudia Bausewein**[4], **Friedemann Nauck**[1], on behalf of the PallPan Study Group[¶]

1 Department of Palliative Medicine, University Medical Center, Georg August University Göttingen, Göttingen, Germany, 2 Department of Palliative Care, Jena University Hospital, Jena, Germany, 3 Department of Palliative Medicine, Faculty of Medicine and University Hospital, University of Cologne, Cologne, Germany, 4 Department of Palliative Medicine, Munich University Hospital, Munich, Germany

☯ These authors contributed equally to this work.
¶ Membership of the PallPan Study Group is listed in the Acknowledgments.
* Maximiliane.jansky@med.uni-goettingen.de

**Data Availability Statement:** Data cannot be shared publicly because of limited anonymity. Data are available from the Department for Palliative

## Abstract

### Background

The COVID-19 pandemic has affected health care systems worldwide. Multidisciplinary teams provide specialist palliative home care (SPHC) for patients with incurable, severe, progressive diseases. These patients are at the same time at high risk, if infected, highly constricted by containment measures, and dependent on support.

### Aim

To explore i) how German SPHC teams were affected by the pandemic during the first wave, ii) which challenges they faced, and iii) which strategies helped to handle the consequences of the COVID-19 pandemic for providing good SPHC.

### Method

Four focus groups (with representatives of 18 SPHC teams) and five guided interviews with stakeholders were conducted and analysed using qualitative content analysis.

### Results

Seven key categories emerged from the data. A category in the background describes dependence on organizational characteristics (e.g. sponsorship), which varied by regional factors. Information management was a challenge to SPHC teams, as they had to collect, interpret and adapt, and disseminate information. They reported a shift in patient care because of the COVID-19 pandemic, due to restricted home visits, visitation ban in nursing

Medicine at the University Medical Center Göttingen (contact via palliativ.forschung@med. uni-goettingen.de) for researchers who meet the criteria for access to confidential data.

**Funding:** PallPan is funded within the Network University Medicine by the German Ministry of Education and Research (Förderkennzeichen 01KX2021). https://www.netzwerk-universitaetsmedizin.de/projekte The funders had no role in study design, data collection and analysis, decision to publish, or preparation of the manuscript.

**Competing interests:** The authors have declared that no competing interests exist.

homes, and difficulties for hospital, hospice and nursing home admissions. Measures to reduce risk of infection impeded teamwork. Teams relied upon their local networks in crisis management, but felt often overlooked by local health authorities. Their respective SPHC state associations supported them in information management and exchange.

## Discussion

The pandemic has severely impacted home care for especially vulnerable seriously ill and dying people. A good network with local health providers and authorities could help to harmonize local regulations and ensure quality care for all patient groups. SPHC teams could play an important role in caring for palliative patients with COVID-19 who are not admitted to a hospital due to preferences or resources.

## Background

The ongoing COVID-19 pandemic has affected all aspects of life, and especially the health care system and those working in it. Palliative care aims to relieve suffering and improve quality of life of patients with incurable, severe, progressive disease and their families, mainly, but not strictly limited to the end of life [1–3]. It can be provided both on primary and specialist care level, and as inpatient or outpatient care, including home care [2].

In Germany, the measures to contain the pandemic focused on preventing infections and ensuring that enough resources are available to treat COVID-19-patients. The strategy of the German Government was to "flatten the curve" and prevent a collapse of the health care system, especially intensive care, as observed in other countries. This led to measures that also severely affected health care for patients not infected with COVID-19. During the first pandemic wave, hospitals restricted their admissions, had to keep beds free, and implemented rigorous visitation bans. People had to restrict private contacts, and nursing homes closed for visitors [4]. All these measures may have severely impacted the situation of seriously ill and dying patients and their families [5, 6]. Patients in hospitals and nursing homes died alone, and patients at home did not receive appropriate care due to a lack of personal protective equipment (PPE) for general practitioners, nursing services and other health care professionals. This caused great distress for patients and their caregivers [7, 8].

Most patients with need for palliative care at home receive general palliative care from general practitioners and nursing services. For patients with complex needs, which require the input of specialists in palliative medicine and nursing, specialist palliative home care teams (SPHC) can provide multi-professional care that ensures quality of life and helps patients to stay at home until death [2, 9, 10]. These teams cooperate with general palliative care as well as other health care professionals and palliative care providers, like volunteer hospice services, palliative care units, inpatient hospices, pharmacies, physiotherapists, psychologists, and social workers. SPHC teams can provide care at home, in nursing homes, in homes for people with disabilities, and in inpatient hospices. Unlike inpatient palliative care units and hospices, which had to ban visitors and were sometimes even closed to make room for COVID-19 patients, the care actions of palliative home care teams were not officially restricted in Germany. Nevertheless, contact restrictions and the effects on their network partners may have posed a high burden on SPHC teams. This may have led to difficulties in providing adequate, high quality palliative care for this especially vulnerable patient group, and the impact the pandemic and the containment measures had on SPHC teams should therefore be evaluated.

## Aim

In this study, we aim to explore i) how German SPHC teams were affected by the pandemic during the first wave, ii) which challenges they faced, and iii) which strategies helped to handle the consequences of the COVID-19 pandemic for providing good SPHC.

## Materials and methods

This study is part of the PallPan study, that aims to develop a national strategy for care of seriously ill patients and their relatives during pandemic times in Germany.

Our study part focused on SPHC and volunteer hospice services. We used both qualitative and quantitative research methods to assess the subject. The results of the qualitative study part reported here were used to inform the development of a questionnaire, which was distributed to SPHC teams nationwide. Results from the questionnaire as well as results for volunteer hospice services will be reported elsewhere. To ensure a high scientific standard and transparency, the reporting of this study will follow the "Consolidated criteria for reporting qualitative research"(COREQ) Checklist [11].

### Recruitment

We used online focus groups to explore experiences of SPHC teams. Additionally, we interviewed stakeholders from state associations for SPHC, which exist in almost every federal state in Germany, and which represent the interests of SPHC teams.

Participants were recruited from a pool of SHPC teams (n = 186) that had contributed data about their organisational characteristics to a previous study [9]. We had used the organisational characteristics to construct models of SPHC teams using Latent Class Analysis (small, independent teams; large network teams; small network teams; hospital-based teams). For the current study, we also divided teams into 6 classes depending on the incidence of COVID-19 cases per 100.000 inhabitants in their administrative district (Table 1). We could only identify few teams in the highest classes, and therefor merged class 4, 5 and 6 to create a broader "high incidence"-class. Teams were recruited according to their incidence resulting in four focus groups (high, rather high, rather low, low incidence). In the groups, we varied the team models to create heterogeneity regarding structural characteristics in the groups.

Teams were contacted via phone and informed about the study, its aims and conduct. Participants from teams should have coordinating or leading function and a good overview over teams' activities. If teams were interested to participate in the focus groups, we sent them the study information including data management procedures via mail or telefax. Participants received an informed consent form and sent it back via mail or telefax.

**Table 1. Incidence classes and identified/participating teams.**

| Class | Incidence per 100.000 inhabitants[1] | Number of identified teams | Number of participating teams |
|---|---|---|---|
| 6 | 935 and more | 1 | 4 |
| 5 | 641 up to 934 | 7 | |
| 4 | 421 up to 640 | 15 | |
| 3 | 290 up to 420 | 31 | 4 |
| 2 | 169 up to 289 | 60 | 5 |
| 1 | 0 up to 168 | 72 | 5 |

[1]The classes were defined as used by the Robert-Koch-Institut (the national Public Health Institute in Germany), in their dashboard in August 2020 (https://experience.arcgis.com/experience/478220a4c454480e823b17327b2bf1d4/page/page_1/).

Furthermore, we recruited stakeholders from federal states where we could not recruit SPHC providers, and those highly engaged in a nationwide task force. Stakeholders were contacted via phone, and received a study information including data management procedures, and an informed consent form via mail. They returned the informed consent via mail or telefax.

## Data collection

We prepared a focus group guide that contained the following questions:

How did the pandemic affect the work of SPHC teams? Did they face challenges to their work? Were there differences between pandemic phases (first wave in March and April vs. low incidence during summer)?

How did they meet challenges? Which strategies worked best for them? What challenges are they currently facing?

What strategies and solutions would they chose again to deal with a pandemic? Where do they need more support?

The participants received the guide beforehand, to give them the opportunity to discuss the topics with their multi-professional teams. During the focus groups, we used knowledge mapping to create a focus group illustration map [12]. With this method, a researcher records results as a "mind map" which can be discussed with the participants to validate results and ensure that their statements were understood. We discussed the knowledge map with each focus group at the end of the discussion. Three researchers took part in the focus groups. FS moderated the focus groups, MJ made notes and supported the moderation, NR prepared the focus group illustration maps during the focus group. FS, MJ and NR work as researchers in the Department of Palliative Medicine at the University Medical Center Goettingen. FS is a sociologist and public health researcher, with experience in conducting focus groups with health care providers in palliative care. She works in a research project that focuses on SPHC. MJ is a senior researcher and psychologist, she has worked in several research projects that assessed aspects of SPHC and is experienced in conducting qualitative interviews. NR is a geographer and public health researcher. She has experience in conducting (online) focus groups. At the beginning of each focus groups, researchers and participants introduced themselves.

Focus Groups were held online using Big Blue Button, a software that is locally hosted by the universities computing centre, which ensured conformation with data security regulations. Focus groups were recorded with an inbuilt recorder (video and audio), and additionally on a dictaphone (audio only).

For stakeholder interviews, we adapted the focus group guide to the perspective of state associations.

Interviews with stakeholders were conducted by phone (n = 4) and online using Big Blue Button (n = 1) by MJ. They were recorded on a dictaphone (telephone interviews) and with an inbuilt recorder (online interview).

Both focus groups and interviews were transcribed verbatim.

After both study parts (focus groups/interviews and questionnaire) were completed, all participants were invited to an online meeting, in which the results of the study were presented to them, and they were asked to provide feedback.

## Data analysis

Both focus groups and interviews were analysed using qualitative content analysis [13]. Analysis was supported by the software MaxQDA Analytics Pro 2020. Three research questions presented in the focus group guide (pandemic challenges, strategies to handle them, and

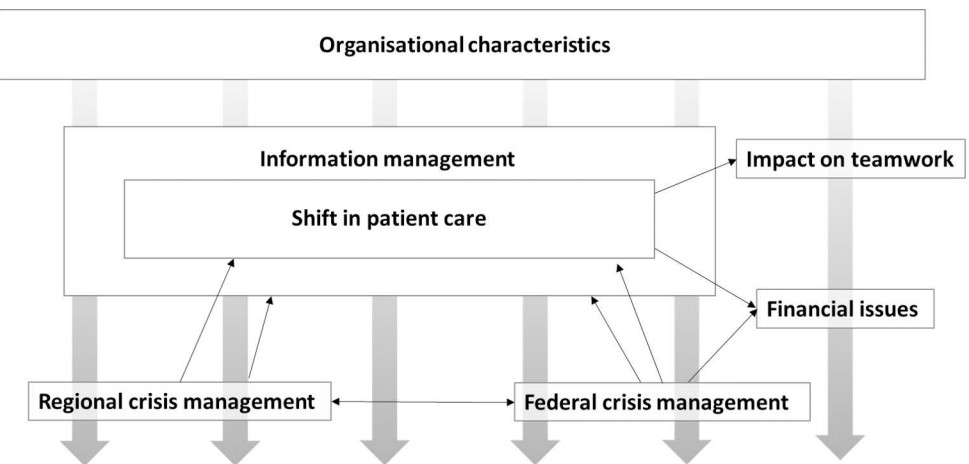

**Fig 1. Key categories of SPHC teams during the COVID-19 pandemic, and their relationships.**

solutions) guided the analysis. First, the focus group transcripts were paraphrased and coded, and categories were formed inductively from the content. Second, categories were combined to form condensed categories. FN and MJ discussed codes and categories until consensus was reached. Then, we used these categories to code the stakeholder interviews. Codes that did not fit into the existing categories were combined to form new categories. Finally, seven core categories were extracted. FS and MJ discussed the final coding schemes. During the coding, various associations and dependencies between categories emerged, which are outlined in Fig 1.

## Ethics

The study received approval from the Research Ethics Committee of the University Clinic Jena (2020-1848-Bef) and the University Medical Center Göttingen (16-8-20 COVID-19).

# Results

## Sample

Between September 15, 2020, and September 29, 2020, four focus groups with 4–5 participants were conducted. In the focus groups "high incidence" and "rather low incidence", two participants from one team took part. Therefore, we had 20 participating staff members, representing 18 teams. Most participants were either leading physicians or coordinating nurses (Table 2). A majority of the participants was female. Teams represented different federal states and all focus groups where mixed regarding federal states, except for the high incidence group in which only Bavarian teams participated. Some of the participants already knew FS or MJ from previous studies. Focus group duration was between 1 hour 34 minutes and 2 hours, including discussion of the focus group illustration maps.

Of the five stakeholders we interviewed, four were female and one was male. They were nurses (3), physicians (1) or executive secretaries (1) who worked as representatives of their respective state association. Stakeholder interview lasted between 56 minutes and 1 hour 18 minutes.

## Key categories

A total of seven key categories emerged from the data. Fig 1 gives an overview of the categories and shows their relationships to each other. To support our interpretation of the data, we provide quotations to illustrate the categories in **Table 3**.

**Table 2. Participant and team characteristics.**

| Characteristics | Participants (n = 20) | Teams (n = 18) |
|---|---|---|
| Professional background | | |
| Physician | 9 | |
| Coordinator or coordinating nurse | 10 | |
| Social worker | 1 | |
| Gender | | |
| Male | 6 | |
| Female | 14 | |
| Federal states | | |
| Baden-Wurttemberg | | 3 |
| Bavaria | | 4 |
| Hessia | | 1 |
| Lower Saxony | | 4 |
| North Rhine-Westphalia | | 1 |
| Rhineland-Palatinate | | 2 |
| Saarland | | 1 |
| Schleswig-Holstein | | 1 |
| Thuringia | | 1 |

**Organisational characteristics.** A key category that influenced all other categories were the organisational characteristics of teams. Teams' experiences differed depending on their sponsorship (e.g. hospital based, independent, or private practice/nursing service based), contract and reimbursement details, and federal state requirements. Other aspects were their involvement in regional structures, and how familiar regional authorities were with their work.

**Information management.** Information management was a key category described by the participants. Team leaders had to manage a rapidly changing "flood of information". Information was mostly perceived as not fitting for this specific field, therefore it had to be collected, interpreted, adapted to SPHC, and passed on to team members, network partners, and patients and their families. While teams used digital communication channels, they also judged this as difficult, especially when communicating with patients and families. Information management was especially challenging if teams served several districts, or if team members lived in a different district, because regulations (e.g. regarding quarantine after contact, ban on visitation in nursing homes) were sometimes different between districts. Team leaders, who acted as the "information hub" had an increased workload as they had to rapidly adapt to new circumstances. They formed task forces, implemented daily communication routines, developed new standards, planned for emergency and crisis situations, and made sure their teams stayed capable of patient care. Teams that are part of a larger organization (e.g. a hospital, a welfare organization) often received already adapted information material, while teams that worked independently had to collect, interpret and adapt information themselves.

Stakeholders from SPHC state associations tried to support teams regarding collection, adaption and dissemination of information. They used their contacts to the state ministries to rapidly collect and discuss new information, and implemented platforms for SPHC teams to exchange information, like weekly video calls.

**Shift in patient care.** Due to the rapidly changing regulations, participants reported a shift in patient care. It was challenging to provide quality care during a pandemic while

**Table 3. Quotes and categories.**

| Quote | Categories |
|---|---|
| *"We are sponsored by a non-profit association, so we had no support by the county or by hospitals. We had to pay for everything ourselves, and this was really difficult in the beginning, the internet was sold out of single-use products."*(FG III, IP 1) | organisational characteristics |
| | financial issues |
| *"Collecting information was difficult, and then to sort out those adequate for our team, this was a main problem in the first days, to implement the new regulations. And then to find out: where do I get my material from, who is my contact partner, who is responsible, who is authorized, whom do I have to listen to, and where can I say 'This is right now not so important to me'."* (FG I, IP 3) | information management |
| | regional crisis management |
| *"We didn't feel SPHC was adequate, because we had to think long and hard if a visit was necessary."*(FG I, IP 5) | shift in patient care |
| *"(the team) was divided in a knee-jerk-action, as a palliative care team we are complete, and then we split in the middle, and this was really how it felt."*(FG II, IP5) | shift in patient care, teamwork |
| *"we were very much on our own, there was no support on the municipal level, not from political bodies up to the federal level. They said, 'they are part of an independent organisation, so we didn't consider them, and we don't know how to provide them with material.' This was a cost factor, which we can at the moment estimate at 67,000 €, for additional material."* (FG II, IP 5) | organisational characteristics |
| | financial issues |
| | regional crisis management |
| | federal crisis management |
| *"we had the same problems with the health office (…) I don't know how often I called with colleagues in the beginning who said: "(…) we are not responsible and this is a special case." (…) you feel like you don't belong to standard care, no one feels responsible, but we still saw the need."* (FG III, IP2) | regional crisis management |
| *"The crisis shows that it is good to have a state association, because in such a difficult situation, the state needs contact partners, they can't contact each team individually. Then it is very helpful to have a state association that organises itself (…) you know who you must call to get a reaction, and all these tasks can't be done by the state, they simply need a contact partner, they have other concerns."* (stakeholders II) | federal crisis management |
| *"In the beginning, I had some difficulties with the abundance, there was something (information) from the city, from the state, from the hospice and palliative care association, from the professional association. And my task was to decide what do we take from whom, what is the overlap, where are things different, but I had the feeling that what came from the hospice and palliative care association and the professional associations was good. The state regulations I have to read three times, but I felt well supported and informed, I rather had the problem that I hoped our wording would be understood by all staff members. I was very alarmed by how different things were understood."* (FG I, IP 1) | information management |
| | regional crisis management |
| | federal crisis management |
| | team work |

planning and implementing infection control measures to protect their patients and team members from infection. However, preparing to manage patients with COVID-19 infection also required organizational skills. Daily routines were changed: if possible, teams were separated to minimize contacts and to ensure that a team would be available, if quarantine measures had to be taken. Home visits were reduced, most contacts were held by phone, and patients were only visited at home if absolutely necessary. At the same time, teams saw home visits as essential in specialized palliative home care. When possible, they took protective measures (e.g. PPE) to enable home visits. Patients were screened for common COVID-19 symptoms before visits. Teams also implemented disinfection routines, work from home and mobile documentation. They emphasized the heightened need for advance care planning in a pandemic, to avoid unwanted hospital admissions especially for nursing home residents.

Teams had different access to resources: staff planning was complicated by home schooling, quarantine measures and separated teams. Teams that were part of a larger organisation, like a hospital, also faced staff resource relocation to other units. Access to protective equipment was difficult, especially for teams that are not part of a larger organisation. These teams had to

organise PPE themselves while there was considerable shortage on the market. Most teams had difficulties to organise tests for COVID-19 for themselves and their patients. Testing by teams was only possible, if other structures (e.g. hospital with testing infrastructure, family practice which was eligible for testing) were involved.

Teams were also indirectly affected by challenges their network partners experienced. It was difficult to transfer patients to nursing homes or hospices, and patients were reluctant regarding admissions to hospitals. Nursing homes sometimes denied access to patients. General practitioners and ambulatory nursing services experienced an overload or were not available due to a lack of PPE. Volunteer hospice services did not or were not allowed to visit patients. Other supporting therapists, like physiotherapists, were restricted as well. Some teams reported that they had more patients than usual due to these network repercussions, while others reported the opposite.

Teams felt that patients and their families suffered because of the pandemic and its consequences. They were afraid of infection, felt isolated, tried to avoid hospital admissions, and experienced a lack of care provision. Caregivers were perceived as highly burdened. Grief counselling was seen as especially difficult.

**Teamwork.** The shifts in patient care influenced the teamwork of SPHC teams. Their internal communication suffered due to contact restrictions and separation of team members. Meetings were cancelled or held online, and clinical supervision was also cancelled. Team members experienced high emotional burden heightened by fear of infection, which could not be addressed as a team. The measures against the pandemic sometimes caused team conflicts and lead to negotiation processes, in which team leaders had to mediate and sometimes push measures through.

**Financial issues.** The shifts in patient care also led to financial issues: Some teams, especially small, independent teams, struggled with the financial burden of purchasing PPE, which was not reimbursed by health insurances. When contacted, health insurances didn't prioritize decisions on reimbursement for specialist palliative home care. Some teams also had financial losses because their reimbursement plan included fees for home visits, but not for phone calls. These financial issues led to additional uncertainty in teams.

**Regional crisis management.** Regional crisis management was an important factor that influenced the shift in patient care as well as information management. Regional networks that included local health authorities, hospitals, nursing homes, physicians, and palliative home care were able to develop and implement coherent regional regulations. In other regions, every nursing home had its own regulations regarding SPHC teams' access to nursing home residents. Some nursing homes allowed only general practitioners and excluded SPHC teams. Contact and communication between SPHC teams and local health authorities was very heterogeneous, some teams had no contact and couldn't reach local authorities, others felt well supported. Teams also cooperated with other SPHC teams that served in their own or neighbouring areas, e.g. in a large city all SPHC teams cooperated and assigned nursing homes to each team.

**Federal crisis management.** Federal crisis management also influenced both shift in patient care and information management. Every federal state enacted their own regulations to control the pandemic. State associations for SPHC served as contacts for the state ministries. They had varying tasks: they mediated between politics and teams, collected, assessed, and disseminated information, they intervened when teams had difficulties with their local health authorities (e.g. regarding issuing PPE), and they encouraged and facilitated strategy and information exchange between teams in their area. The national association for SPHC also implemented a regular information exchange between state associations.

## Discussion

The ongoing COVID-19 pandemic had various consequences for SPHC teams in Germany. Our study is the first to provide an insight how SPHC was affected by the pandemic, and how SPHC teams responded to it.

SPHC teams experienced difficulties in information management, as information was mostly not suitable for SPHC. This was perceived as a high workload, because information and regulations changed rapidly. With changing regulations patient care and team routines had to be adapted as well. SPHC teams reduced home visits but felt that this was not optimal for SPHC. They lacked a standard for assessing symptoms and patient status via phone. Greenhalgh et al. developed a guide to assess COVID-19 patients status and symptoms via phone or video consultation [14], which could be adapted to SPHC. While SPHC teams in Germany were sometimes reluctant to use telemedicine, palliative care providers in other countries report positive experiences with these measures [15]. Infrastructure in German health care was not prepared to widely implement telehealth. Telemedicine should not replace personal contact in palliative care under normal circumstances, but its additional use could also prove beneficial to patients and providers [16].

While teams changed their routines to protect their patients as well as their staff, and to stay capable of providing care to their vulnerable patients, they also experienced challenges from the measures their network partners took. Some nursing homes allowed access only for one GP and denied the SPHC teams access to their patients [17]. The regulations on visitors in nursing homes, which often differed from institution to institution were described as very challenging. One example demonstrated that this could be prevented by a regional task force which included all relevant health providers and local health authorities as well as scientists.

Teams felt that their patients experienced a lack of care due to a lack of PPE as well as an overload for GPs and nursing services. Schoenmakers and colleagues reported that if a shortage of home care nurses happens because of the pandemic, informal caregivers could be educated to administer drugs and perform other measures that are usually provided by nurses [16]. SPHC always includes families in the care process which could be especially beneficial during a pandemic.

Teamwork is an integral part of palliative care. The contact restrictions led to a separation of team members, forcing them to work alone. Team members felt isolated, missed the informal communication with other team members, and experienced emotional burden [17]. Team leaders offered emotional support and tried to implement measures to enable informal communication, like daily video conferences that had no specific agenda. Some allowed team meetings during summer outside or in large rooms with the possibility to air.

SPHC teams often mentioned that they felt left alone and overlooked, both by federal, state and local authorities. While they themselves felt that they contributed important work during the pandemic and were willing to care for dying COVID-19 patients at home and in nursing homes, they also experienced that their work was not valued. Local, state and federal health authorities were often not aware of SPHC structures, which is reflected by their difficulty to organize PPE and tests for COVID-19 [17]. Personal contact and home visits are an important part of SPHC and limited access to PPE and tests was hence problematic.

SPHC teams were supported by their state associations, which held contact with the ministries and federal associations, and sometimes even intervened when teams had problems with their local authorities. This was seen as extremely helpful by the teams, and they benefited from the exchange with other SPCH teams.

Since our data assessment, SPHC teams have gained more prominence in the public health discourse, they are for example explicitly noted in the German vaccination prioritisation [18].

The pandemic situation changes rapidly, therefore our study provides only insight for a specific timeframe. There were no differences between the focus groups with low and the groups with high incidence. The focus groups were conducted after the first pandemic wave in Germany, therefore we had a rather low incidence, and teams had only little experience with acute COVID-19 -infection among staff and almost no experience regarding COVID-19 infection in patients. This might have changed during the second wave which started shortly after we conducted the focus groups. This was covered by the questionnaire that was developed based on the focus group results.

Transferability of the results to other pandemics is not clear. During other pandemics, specifics of the respective virus, like case-fatality rate, symptoms and contagiousness need to be taken into account.

While the data are collected in Germany and within its health care system, the results may be translatable to other countries, as international research shows that hospice and palliative care in other countries faced similar challenges [17, 19].

## Limitations

Because of the contact restrictions, but also because of the limited time frame of the study, we conducted the focus groups online as video conferences. Therefore, non-verbal communication was limited [20]. While this potentially restricted the groups' interaction, we judged this as acceptable as we were more interested in their experiences and less interested in group dynamics. Using online focus groups allowed us to rapidly recruit participants. It also allowed us to recruit participants who are geographically dispersed and have limited time resources which would have kept them from attending face-to-face focus groups. Due to the pandemic, all participants were familiar with video-conferencing tools [21].

Our focus groups were rather small (4–5 participants), due to the limited recruitment time, and the high workload of SPHC teams. This may have limited information output. On the other hand, it allowed us to give every participant enough opportunity to contribute and to explore the SPHC teams experiences more deeply.

## Conclusion

The COVID-19 pandemic has impacted SPHC in various ways. Due to their unique and heterogeneous structure, teams were often overseen by health authorities. Critically ill COVID-19 patients are in some cases not admitted to hospitals because of their personal preferences, but also because of limited resources in hospitals. SPHC teams could be the key provider to ensure good care for these patients, at home or in nursing homes. Additionally, SPHC patients are highly vulnerable, and especially impacted by the pandemic measures, because contact restrictions and fear of infection severely limit their ability to be with loved ones during their remaining time. Although visits to a dying person were officially excluded from contact restrictions in Germany, this was not always adequately communicated and put into practice [8]. Specialist palliative care, including home care, should receive more attention and be involved in planning and regulating health care during a pandemic. The PallPan project has developed a national strategy for palliative care of severely ill and dying people and their relatives in pandemics [22], which contains recommendations for general and specialist care providers.

## Acknowledgments

We would like to thank all study participants for contributing their time and experiences. We although thank the PallPan study group:

Munich University Hospital, Department of Palliative Medicine, Germany: Daniela Gesell, MSc, Eva Lehmann, MSc, Sonja Gauder, Dr. Farina Hodiamont, Jerri Bazata, MA, Sophie Meesters, MPH, Nathalie Berges, Marie Wallner

University of Cologne, Faculty of Medicine and University Hospital, Department of Palliative Medicine, Germany: Prof. Dr. Raymond Voltz, Dr. Dr. Julia Strupp, Charlotte Leisse, Berenike Pauli, MSc, Karlotta Schlösser, MSc, Dr. Anne Pralong, Buket Dilara Cinar

University of Cologne, Faculty of Medicine and University Hospital, Department I of Internal Medicine: Prof. Dr. Norma Jung

Medical Center–University of Freiburg, Department of Palliative Medicine: Prof. Dr. Gerhild Becker, PD Dr. Christopher Böhlke, MSc

Rostock University Medical Center, Department III (Hematology, Oncology, Palliative Medicine): Prof. Dr. Christian Junghanß, Dr. Ursula Kriesen

Düsseldorf University Hospital, Interdisciplinary Center for Palliative Medicine: Dr. Martin Neukirchen, Dr. Jacqueline Schwartz, Manuela Schallenburger, MSc, Marie Christine Reuters, MSc

University Medical Center Erlangen, Department of Palliative Medicine: Prof. Dr. Christoph Ostgathe, Dr. Dr. Maria Heckel, Isabell Klinger, Sophie Shahda, Clarisse Kugler, Beatrice Wahlen

University Hospital Bonn, Department of Palliative Medicine: Prof. Dr. Lukas Radbruch, Dr. Birgit Jaspers, Dr. Gülay Ates, Katja Maus, MA

University Hospital RWTH Aachen, Department of Palliative Medicine: Prof. Dr. Roman Rolke, Norbert Krumm, MSc

Hannover Medical School, Institute for General Practice and Palliative Care: Prof. Dr. Nils Schneider, Prof. Dr. Stephanie Stiel, Jannik Tielker, Jan Weber

University Hospital Würzburg, Interdisciplinary Center, Palliative Medicine: Prof. Dr. Birgitt van Oorschot, Dr. Carmen Roch, Marius Fischer, Anke Ziegaus, Liane Werner

University Medical Center Hamburg-Eppendorf, Department of Oncology, Hematology and BMT, Palliative Care Unit: Prof. Dr. Karin Oechsle, Dr. Christina Gerlach, MSc, Dipl. Soz. Anneke Ullrich

## Author Contributions

**Conceptualization:** Maximiliane Jansky, Cordula Gebel, Ulrich Wedding, Friedemann Nauck.

**Data curation:** Maximiliane Jansky, Franziska Schade, Nicola Rieder, Danica Lohrmann.

**Formal analysis:** Maximiliane Jansky, Franziska Schade, Nicola Rieder, Danica Lohrmann.

**Funding acquisition:** Maximiliane Jansky, Cordula Gebel, Ulrich Wedding, Steffen T. Simon, Claudia Bausewein, Friedemann Nauck.

**Investigation:** Maximiliane Jansky, Franziska Schade, Nicola Rieder.

**Methodology:** Maximiliane Jansky, Franziska Schade, Lars Kloppenburg.

**Project administration:** Maximiliane Jansky.

**Supervision:** Ulrich Wedding, Steffen T. Simon, Claudia Bausewein, Friedemann Nauck.

**Validation:** Friedemann Nauck.

**Writing – original draft:** Maximiliane Jansky, Franziska Schade.

**Writing – review & editing:** Maximiliane Jansky, Franziska Schade, Nicola Rieder, Danica Lohrmann, Cordula Gebel, Lars Kloppenburg, Ulrich Wedding, Steffen T. Simon, Claudia Bausewein, Friedemann Nauck.

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
