## [Decision Letter · Decision Letter 0]

14 Sep 2021

PONE-D-21-11640‚It felt like a black hole, great uncertainty, but we have to take care for our patients’ – Qualitative findings on the effects of the COVID-19 pandemic on specialist palliative home carePLOS ONE

Dear Dr. Jansky,

Thank you for submitting your manuscript to PLOS ONE. After careful consideration, we feel that it has merit but does not fully meet PLOS ONE’s publication criteria as it currently stands. Therefore, we invite you to submit a revised version of the manuscript that addresses the points raised during the review process.

This is an important article that is likely to have relevance internationally. Two reviews have been completed, please respond to the comments from the reviewers. For reviewer 2, who suggest the dates of the focus groups should be in the methods, I will be guided by the authors as to their preference here as it is my feeling it is common to have this in results, as you have done. However, I would also request that: 1) the authors perform a very detailed proof read, use consistent language for COVID-19 and identify and remove any errors in the paper identified by reviewers;2) PallPan is given context throughout, this is in the abstract but the relevance to the study is not clear. Is it needed in the abstract? Please ensure it is defined and clear for an audience not familiar with this project,3) For the results, would the authors consider using subheaders? There are not many quotes presented, please consider whether the findings are adequately supported with data.4) I found figure 1 to be confusing and hard to read. Please either provide a clearer version and clarify how it contributes to interpretation of the findings, or remove. 

We look forward to receiving your revised manuscript.

Kind regards,

Anna Ugalde, PhD

Academic Editor

PLOS ONE

Journal Requirements:

Additional Editor Comments (if provided):

Thank you for this submission to Plos One. This is an important article that is likely to have relevance internationally. Two reviews have been completed, please respond to the comments from the reviewers. For reviewer 2, who suggest the dates of the focus groups should be in the methods, I will be guided by the authors as to their preference here as it is my feeling it is common to have this in results as they have done. However, I would also request that: 1) the authors perform a very detailed proof read, use consistent language for COVID-19 and identify and remove any errors in the paper identified by reviewers; 2) PallPan is given context throughout, this is in the abstract but the relevance to the study is not clear. Is it needed in the abstract? Please ensure it is defined and clear for an audience not familiar with this project, 3) For the results, would the authors consider using subheaders? There are not many quotes presented, please consider whether the findings are adequately supported with data. 4) I found figure 1 to be confusing and hard to read. Please either provide a clearer version and clarify how it contributes to interpretation of the findings, or remove.

Reviewers' comments:

Reviewer's Responses to Questions

**Comments to the Author**

1. Is the manuscript technically sound, and do the data support the conclusions?

Reviewer #1: Partly

Reviewer #2: Partly

2. Has the statistical analysis been performed appropriately and rigorously? 

Reviewer #1: No

Reviewer #2: N/A

3. Have the authors made all data underlying the findings in their manuscript fully available?

Reviewer #1: No

Reviewer #2: Yes

4. Is the manuscript presented in an intelligible fashion and written in standard English?

Reviewer #1: Yes

Reviewer #2: Yes

5. Review Comments to the Author

Reviewer #1: Thank you for the opportunity to review your manuscript. I am sure that the experiences of German specialist palliative home care teams mirrored the experiences of most home-based palliative care teams during the first and subsequent waves of the COVID-19 pandemic.

General comments:

-Either consistently refering to the virus as COVID-19 or SARS-CoV-2 throughout rather than using the terms interchangeably. My recommendation would be to stick with COVID_19 as it is in common usage and recognisable to all sectors of the community.

- Careful editing will eliminate the syntax and grammar errors throughout- there are some challenges to both meaning and fluency but they are minimal.

-Review use of possessive punctuation, for example, it's.

-Review use of brackets and abbreviations- they interfere with fluency. Review use of contractions - for example L238.

- Review sentence structure of some of your longer sentences - some would benefit simply from use of semi-colons or divided into shorter sentences - for example L165-169.

- Urgently review use of "quotation marks" it is unclear if you are indicating verbatim quotations from references or direct quotes from the focus group participants or stakeholder interviews.

-L118-125 reads rather like resumes and does not belong in the body of the manuscript.

Tables and figures:

Table 1: suggest you simplify and remove the focus group composition column and either discuss the composition or include another table. The link to the website you referred to below the table takes you to a blank page with a heading but nothing else.

Figure 1: does not support the inter-connections you discussed in your manuscript- consider re-imagining it to make these connections reflect what you claimed in your results. You might also consider use of an infographic to improve the visual impact of the diagram.

Method: L 70 what do you mean by consent?

You claim to have used qualitative content analysis Philipp (2010- note there is a 2015 version available) but little detail of the process is offered.

Results: are reported with little original (or translated) text offered - this makes it challenging to discern if the interpretations you made were supported by the data you collected. It would strengthen your analysis considerably to support it with quotes from at least a couple of different participants/stakeholders otherwise it is more of a summary than a qualitative analysis.

Consider more clearly outlining the recommendations for future practice as these will be of interest to providers.

Reviewer #2: I read this paper with interest and thank the authors for sharing their work in this important area. I offer the following comments as feedback to help improve its potential contribution to the field/lliterature:

* Please review and correct punctuation used in the title - it appears that a comma is used at the beginning rather than an opening quotation/speech mark (this should correspond with the quotation/speech mark that is correctly used at the end of the quotation).

* Abstract - it seems inconsistent that there is no heading for the first (background/introduction) part of the abstract, but then there are headings for Aim/Method/Results/Discussion. Suggest adding a heading for the first section for consistency.

* use of COREQ reporting guidelines/checklist - I could not see any mention of COREQ checklist Item 28 - whether the study participants provided feedback on the findings? If not, were they offered a reasonable opportunity to do so? Please address this as per COREQ.

* Results - Is "Between September 15, 2020 and September 29, 2020, four focus groups with 4-5 participants were conducted" really a result of the study? I would suggest that the conduct of these FGs is more accurately a research method of data collection. As such, I think this sentence should be relocated to the appropriate ,methods/data collection section - rather than being reported here as a 'result' of the study.

* Limitations - Reference #21 - cited in support of using online FGs: I note this supporting reference comes from the 'Journal of Advertising' wonder why this is considered appropriate? I suggest that you could find a much more relevant source to cite here in this (health research) context. For example:

https://www.ncbi.nlm.nih.gov/pmc/articles/PMC7550163/

6. PLOS authors have the option to publish the peer review history of their article (what does this mean?). If published, this will include your full peer review and any attached files.

Reviewer #1: **Yes: **Katrina Recoche

Reviewer #2: No

---

## [Author Response · Author response to Decision Letter 0]

23 Oct 2021

Thank you for giving us the chance to resubmit our manuscript “‘It felt like a black hole, great uncertainty, but we have to take care for our patients’ – Qualitative findings on the effects of the COVID-19 pandemic on specialist palliative home care" after we received very helpful feedback from our peer reviewers.

We hope to have answered all questions that arose in our point-by-point reply. We are very thankful for the helpful suggestions of the reviewers, which improved the manuscript significantly.

Please let us know if there are any other points to address. 

PallPan is given context throughout, this is in the abstract but the relevance to the study is not clear. Is it needed in the abstract? Please ensure it is defined and clear for an audience not familiar with this project, Thank you. We have deleted the reference to the PallPan project in the abstract.

For the results, would the authors consider using subheaders? There are not many quotes presented, please consider whether the findings are adequately supported with data Thank you for this suggestion. We added subheaders in the results section. We also added a table that contains quotes to support our data analysis. 

I found figure 1 to be confusing and hard to read. Please either provide a clearer version and clarify how it contributes to interpretation of the findings, or remove. Thank you. We had some technical difficulties when uploading the figure. We have simplified the figure and also made the interconnections in the text more prominent. We believe that the figure is helpful to provide an overview of the key categories and their connections.

Either consistently refering to the virus as COVID-19 or SARS-CoV-2 throughout rather than using the terms interchangeably. My recommendation would be to stick with COVID_19 as it is in common usage and recognisable to all sectors of the community.

 Thank you for this comment! We have changed all mentions of SARS-CoV-2 to COVID-19.

Careful editing will eliminate the syntax and grammar errors throughout- there are some challenges to both meaning and fluency but they are minimal. Thank you! We have reviewed the text again and hope to have eliminated all the challenges.

Review use of possessive punctuation, for example, it's. Thank you! We have reviewed the possessive punctuation and hope that we have corrected all errors.

Review use of brackets and abbreviations- they interfere with fluency. Review use of contractions - for example L238. Thank you for this comment. We have reviewed the use of brackets and abbreviations, and corrected them, where possible. 

Urgently review use of "quotation marks" it is unclear if you are indicating verbatim quotations from references or direct quotes from the focus group participants or stakeholder interviews.

 Thank you! 

L118-125 reads rather like resumes and does not belong in the body of the manuscript. Thank you for this comment. We included this part to adhere to the COREQ checklist (items 1 to 8) and therefore would keep it in the manuscript. 

Table 1: suggest you simplify and remove the focus group composition column and either discuss the composition or include another table. The link to the website you referred to below the table takes you to a blank page with a heading but nothing else.

 Thank you! We have added another table with the participants’ characteristics.

Figure 1: does not support the inter-connections you discussed in your manuscript- consider re-imagining it to make these connections reflect what you claimed in your results. You might also consider use of an infographic to improve the visual impact of the diagram.

 Thank you. We had some technical difficulties when uploading the figure. We have simplified the figure and also made the interconnections in the text more prominent. We believe that the figure is helpful to provide an overview of the key categories and their connections.

Method: L 70 what do you mean by consent? One part of the PallPan study was to conduct a consensus process for the developed strategy and recommendations. As this is not part of the reported study part, we have deleted this part of the sentence.

You claim to have used qualitative content analysis Philipp (2010- note there is a 2015 version available) but little detail of the process is offered. Thank you! We have changed the part on data analysis to include more details on the data analysis process. We have also updated the version of the referenced book.

Results: are reported with little original (or translated) text offered - this makes it challenging to discern if the interpretations you made were supported by the data you collected. It would strengthen your analysis considerably to support it with quotes from at least a couple of different participants/stakeholders otherwise it is more of a summary than a qualitative analysis.

 Thank you for the suggestion to add original text. We have added a quote for every category to support our interpretation. Quotes are presented in form of a table, because quotes were often associated with more than one category.

Consider more clearly outlining the recommendations for future practice as these will be of interest to providers. Thank you. We have added another reference to the recommendations that were developed in the PallPan-Project.

* Please review and correct punctuation used in the title - it appears that a comma is used at the beginning rather than an opening quotation/speech mark (this should correspond with the quotation/speech mark that is correctly used at the end of the quotation). Thank you, we have deleted the comma and used a quotation mark.

* Abstract - it seems inconsistent that there is no heading for the first (background/introduction) part of the abstract, but then there are headings for Aim/Method/Results/Discussion. Suggest adding a heading for the first section for consistency. Thank you. We have added a heading to this section

* use of COREQ reporting guidelines/checklist - I could not see any mention of COREQ checklist Item 28 - whether the study participants provided feedback on the findings? If not, were they offered a reasonable opportunity to do so? Please address this as per COREQ.

 Thank you for raising this important point! The teams were invited to an online meeting, where the results of the study were presented to them, and they were asked to provide feedback. 

* Results - Is "Between September 15, 2020 and September 29, 2020, four focus groups with 4-5 participants were conducted" really a result of the study? I would suggest that the conduct of these FGs is more accurately a research method of data collection. As such, I think this sentence should be relocated to the appropriate ,methods/data collection section - rather than being reported here as a 'result' of the study.

 Thank you for this comment that we carefully considered. We believe it is common to place the time of data collection in the results section. Furthermore, because of the dynamic progression of the pandemic, the timing of the data assessment directly influences the results and discussion. We therefore considered this information as adequately placed in the results section. 

* Limitations - Reference #21 - cited in support of using online FGs: I note this supporting reference comes from the 'Journal of Advertising' wonder why this is considered appropriate? I suggest that you could find a much more relevant source to cite here in this (health research) context. Thank you for providing such an excellent reference. When the manuscript was first written, there was not much information available regarding the use of virtual focus groups. We have added the reference you suggested.

---

## [Editor Report · Decision Letter 1]

17 Nov 2021

'It felt like a black hole, great uncertainty, but we have to take care for our patients’ – Qualitative findings on the effects of the COVID-19 pandemic on specialist palliative home care

PONE-D-21-11640R1

Dear Dr. Jansky,

We’re pleased to inform you that your manuscript has been judged scientifically suitable for publication and will be formally accepted for publication once it meets all outstanding technical requirements.

Kind regards,

Anna Ugalde, PhD

Academic Editor

PLOS ONE
---

## [Editor Report · Acceptance letter]

6 Dec 2021

PONE-D-21-11640R1 

‘It felt like a black hole, great uncertainty, but we have to take care for our patients’ – Qualitative findings on the effects of the COVID-19 pandemic on specialist palliative home care 

Dear Dr. Jansky:

I'm pleased to inform you that your manuscript has been deemed suitable for publication in PLOS ONE. Congratulations! Your manuscript is now with our production department. 

Kind regards, 

on behalf of

Dr. Anna Ugalde 

Academic Editor

PLOS ONE